# Modular Distillation Makes Small Models Think Like Big Ones

## Abstract

Large Language Models (LLMs) have demonstrated exceptional performance in knowledge-sensitive reasoning tasks, but their practical application is still restricted by high computing demand. To address these challenges, we propose a novel modular distillation framework that breaks down knowledge-intensive reasoning tasks into three distinct components: an Analyzer for question decomposition, a Informant for context building, and a Reasoner for step-by-step reasoning inference. Unlike previous distillation methods that focus only on matching final outputs or step-by-step reasoning, our approach introduces a structured pipeline that enables the student model to learn both the analytical and reasoning abilities of the teacher model, while also capturing the teacher's internal knowledge to guide more accurate and informed inference. This architecture improves interpretability, efficiency, and modularity, allowing for independent optimization of subcomponents. Empirical tests on three different benchmarks–OBQA, StrategyQA, and MedQA–show that our framework outperforms monolithic baselines in accuracy and computing efficiency while achieving competitive performance with much smaller models. Our findings demonstrate that smaller language models can do reasoning more efficiently when the whole process is divided into more manageable distinct components. This modular approach offers a practical and transparent alternative to relying on extremely large, resource-intensive models[1].

## 1 Introduction

Large Language Models (LLMs) are showing important capabilities in understanding and generating text, which makes them useful tools for a wide range of applications, from daily conversations to complex reasoning tasks (Vaswani et al., 2017; Brown et al., 2020; Wei et al., 2022; Ouyang et al., 2022; Touvron et al., 2023; DeepSeek-AI et al., 2025). It is a well-known problem that deploying these models to real-world applications often encounters challenges due to their computational cost and latency. At this point, knowledge distillation arises as a viable solution for transferring expertise from a powerful *Teacher* model to a smaller *Student* model (Song et al., 2024; Gu et al., 2024; Mansourian et al., 2025; Tian et al., 2025). Nevertheless, traditional knowledge distillation methods usually focus on a limited subset of the teacher's capabilities. They mainly concentrate on replicating outputs while ignoring critical skills like analyzing, contextual understanding, and creating reasoning trace (Magister et al., 2022).

While traditional knowledge distillation approaches focus on a single teacher-student pair, in this paper we introduce a novel three-module knowledge and reasoning distillation framework. We specifically target knowledge-intensive reasoning tasks such as medical diagnosis or complex open-domain QA–where the primary challenge lies in accurately retrieving and synthesizing domain-specific information, rather than purely calculation. This framework decomposes the knowledge-intensive reasoning process into specialized subtasks, thereby improving both efficiency and interpretability. In other words, our goal is to preserve and transfer the entire range of capabilities of the teacher model by distilling the teacher's underlying reasoning dynamics (decomposition, information, and synthesis), rather than merely trying to mimick its final output.

In our approach, the reasoning pipeline is divided into three different components: *Analyzer*, *Informant*, and *Reasoner*. First, the *Analyzer* decomposes an input query into a set of useful subquestions,

---

[1]We will publish the complete source code and pretrained models upon publication.

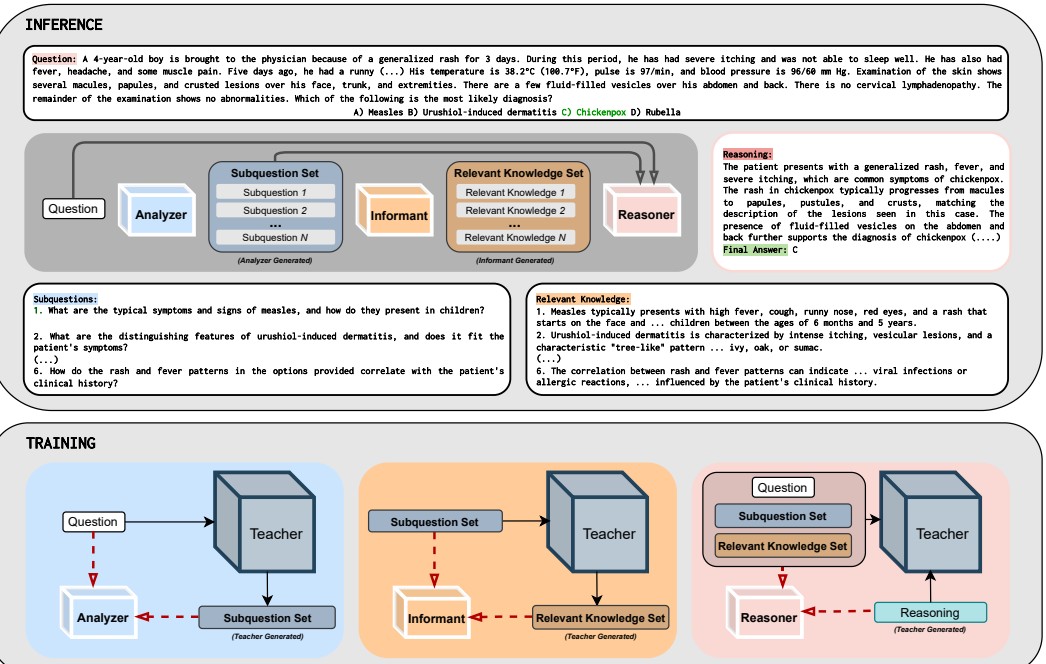

Figure 1: Overview of our modular knowledge distillation framework, showing both inference and training workflows. **Top: Inference Stage** – The system operates in three sequential stages for given query: the *Analyzer* decomposes the question into a set of useful subquestions; the *Informant* generates relevant knowledge for each subquestion and the *Reasoner* integrates the original question, the subquestions, and the associated knowledge to perform reasoning and produce the final answer. **Bottom: Training Stage** – each module is trained independently using supervision from a *Teacher* model. The *Teacher* provides subquestions for training the *Analyzer*, relevant knowledge snippets for the *Informant*, and detailed reasoning traces with final answers for the *Reasoner*. This design enables targeted optimization of each module idenependently during inference.

each targeting a different perspective of the query. Next, the *Informant* addresses these subquestions to construct a comprehensive and well-grounded knowledge base. Finally, the *Reasoner* synthesizes this information to generate a coherent, step-by-step reasoning process that produces the final answer.

This modular design comes with important benefits compared to traditional distillation methods. Dividing the task into parts improves efficiency because each module can be optimized independently. In addition, the modularity offers great architectural adaptability as the specialized modules can now be swapped without retraining the entire core logic. For instance, the *Informant* module could be replaced by a code interpreter for programming tasks or a symbolic calculator for mathematical reasoning, extending the framework's utility beyond textual question answering. The separation of question decomposition, knowledge generation, and reasoning improves interpretability as we can trace how each intermediate step contributes to the final answer. Our experiments demonstrate that this approach not only surpasses the performance of larger monolithic models but also makes the reasoning process fully transparent and traceable. The contributions of this work are below:

- **A novel three-stage distillation framework**: We propose a decomposition-based knowledge and reasoning distillation method that improves both efficiency and interpretability by splitting monolithic bigger model into *Analyzer*, *Informant*, and *Reasoner* modules.

- **Improved question understanding**: The *Analyzer*'s ability to generate subquestions ensures more precise knowledge retrieval and reasoning, particularly for complex queries.

- **Scalability and adaptability**: Each component can be independently fine-tuned or replaced, making the system adaptable to diverse applications without retraining the entire model.

## 2 RELATED WORK

### 2.1 LARGE LANGUAGE MODELS

Large Language Models (LLMs) have demonstrated impressive capabilities across diverse tasks, particularly in applying acquired knowledge to address complex reasoning challenges. Models such as GPT-4 (Achiam et al., 2023), DeepSeek-R1 (Liu et al., 2024), and LLaMA 3 (Dubey et al., 2024) have performed remarkably well on challenging, knowledge-intensive benchmarks, demonstrating their skill in effectively applying acquired knowledge during reasoning. While LLMs continue to improve, their deployment introduces a new challenge forcing users to choose between performance and control (Shanmugarasa et al., 2025; Huang et al., 2025). This challenge mainly comes from the fact that most LLMs can only be accessed through APIs or require a lot of computing power to run. As a result, there is an increasing need for alternative methods that can still make use of LLMs' strengths without needing high computational resources or relying on black-box systems.

### 2.2 KNOWLEDGE DISTILLATION FOR NEURAL NETWORKS

Knowledge Distillation (KD) is a technique that aims to transfer the capabilities of a large, high-performing *Teacher* model to a smaller, more efficient *Student* model (Hinton et al., 2015). In traditional knowledge distillation methods, the aim is to minimize the divergence between the output distributions of the teacher and student, allowing the student to learn not only from ground-truth labels but also from the teacher's soft predictions. This improves the generalization ability of the student model regarding the task.

Since its introduction, Knowledge Distillation has been widely adopted across natural language processing (NLP) tasks, ranging from text classification and question answering to machine translation (Sanh et al., 2019; Jiao et al., 2019; Sun et al., 2020). The core motivation is to reduce model size and computation cost without losing performance. Techniques have evolved to include layer-wise distillation, attention distillation, and hidden-state matching to capture the internal behaviors of the Teacher model (Wang et al., 2020).

With the advance of Large Language Models, Knowledge Distillation becomes a necessary technique for making the deployment of these typically resource-demanding models more accessible to a wider range. However, the complexity of LLM tasks – especially those involving reasoning – introduces new challenges for traditional KD approaches, which are typically designed for simpler output matching.

### 2.3 REASONING DISTILLATION IN LLMS

As large language models are increasingly applied to tasks involving multi-step reasoning, attention is growing not only on their final outputs but also on the reasoning processes behind them. This introduces a different dimension to reasoning distillation: rather than aligning output probabilities alone, the student must also replicate intermediate reasoning steps, such as chain-of-thought (CoT) explanations or sub-question decomposition.

Recent works (Li et al., 2024; Ranaldi & Freitas, 2024; Magister et al., 2022; Hsieh et al., 2023; Yuan et al., 2024; Zhang et al., 2025; Lobo et al., 2024; Liu et al., 2023) have investigated how to distill reasoning processes from teacher LLMs. These studies suggest that guiding smaller models with CoT outputs from a teacher model allows them to achieve performance close to that of the teacher. Even with limited data, small models can effectively learn complex reasoning patterns when trained on carefully selected CoT examples or structured explanations.

Some approaches enhance distillation by integrating external knowledge into the process, where the teacher provides not only the final answer but also provision for the reasoning path (Lee et al., 2024). Another study explores how to combine different forms of distillation signals. For example, rather than transferring only rationales, teacher model provides mixed supervision that alternates between rationales and answers (Li et al., 2023).

Some recent approaches attempt to improve distillation by integrating external knowledge through retrieval systems (Kang et al., 2023; Liao et al., 2024; Lyu et al., 2024; Fu et al., 2023). In these cases, the student model obtains knowledge not from the teacher itself but from external sources,

which may be noisy or inconsistent with the teacher's internal knowledge. Separating reasoning from knowledge creates new challenges, as the student model can depend on the teacher's full understanding during distillation. Some studies have addressed this by suggesting that using the teacher's own knowledge base during training can provide more stable and reliable guidance (Du et al., 2025). Others have highlighted the value of breaking down complex questions into simpler sub-questions during distillation (Wu et al., 2024). This strategy which is known as question decomposition, helps student models better grasp and reproduce multi-step reasoning.

## 3 METHODOLOGY

We propose a modular knowledge distillation framework that factorizes the reasoning process of a large teacher model into three complementary components: the *Analyzer* $\mathcal{A}$, the *Informant* $\mathcal{K}$, and the *Reasoner* $\mathcal{R}$. Unlike traditional distillation, which directly approximates the conditional distribution $P(r_i \mid q_i)$ of teacher reasoning traces $r_i$ given input questions $q_i$, our approach explicitly models latent variables corresponding to intermediate reasoning structure. This enables the student to learn not only final outputs but also the teacher's decomposition and knowledge integration strategies.

### 3.1 PROBLEM SETUP

Given a dataset $\mathcal{D} = \{(q_i, r_i)\}_{i=1}^{N}$ of questions and teacher-generated reasoning traces, we introduce latent subquestions $\{s_{i,j}\}_{j=1}^{n_i}$ and associated knowledge snippets $\{k_{i,j}\}_{j=1}^{n_i}$. The teacher's reasoning pattern can be expressed as

$$P(r_i \mid q_i) = \sum_{\{s_{i,j}\}} \sum_{\{k_{i,j}\}} P(r_i \mid q_i, \{s_{i,j}\}, \{k_{i,j}\}) \, P(\{k_{i,j}\} \mid \{s_{i,j}\}, q_i) \, P(\{s_{i,j}\} \mid q_i). \quad (1)$$

This factorization emphasizes three complementary aspects of reasoning: breaking problems down into subproblems, grounding them through knowledge generation, and integrating the results into a coherent reasoning process.

### 3.2 MODULE DEFINITIONS

The **Analyzer** $\mathcal{A}$ models $P(\{s_{i,j}\} \mid q_i)$ by decomposing each question into conditionally independent subquestions,

$$s_{i,j} \sim P(s_{i,j} \mid q_i; \theta_{\mathcal{A}}), \quad j = 1, \ldots, n_i, \quad (2)$$

where the number of subquestions $n_i$ adapts to the complexity of $q_i$. The **Informant** $\mathcal{K}$ grounds each subquestion by generating contextually relevant knowledge,

$$k_{i,j} \sim P(k_{i,j} \mid s_{i,j}, q_i; \theta_{\mathcal{K}}). \quad (3)$$

Finally, the **Reasoner** $\mathcal{R}$ synthesizes the reasoning and the final answer by conditioning on the original question, its subquestions, and their corresponding knowledge,

$$r_i \sim P(r_i \mid q_i, \{s_{i,j}\}, \{k_{i,j}\}; \theta_{\mathcal{R}}). \quad (4)$$

TRAINING OBJECTIVES

Each module is trained independently using synthetic labels obtained by the teacher. The *Analyzer* is trained to produce teacher-provided subquestions,

$$\mathcal{L}_{\mathcal{A}} = -\sum_{j=1}^{n_i} \log P(s_{i,j} \mid q_i; \theta_{\mathcal{A}}), \quad (5)$$

the *Informant* to generate relevant knowledge,

$$\mathcal{L}_{\mathcal{K}} = -\sum_{j=1}^{n_i} \log P(k_{i,j} \mid s_{i,j}, q_i; \theta_{\mathcal{K}}), \quad (6)$$

and the *Reasoner* to reconstruct the full reasoning trace,

$$\mathcal{L}_{\mathcal{R}} = -\log P(r_i \mid q_i, \{s_{i,j}\}, \{k_{i,j}\}; \theta_{\mathcal{R}}). \quad (7)$$

The overall objective is simply the sum of these losses,

$$\mathcal{L}_{\text{total}} = \mathcal{L}_{\mathcal{A}} + \mathcal{L}_{\mathcal{K}} + \mathcal{L}_{\mathcal{R}}. \quad (8)$$

## 3.3 THEORETICAL GUARANTEES

While traditional knowledge distillation aims to approximate the teacher's conditional distribution $P_T(r|q)$ directly, our framework relies on the hypothesis that the reasoning process is compositional. In this section, we analyze why decomposing the distillation process into *Analyzer* ($\mathcal{A}$), *Informant* ($\mathcal{K}$), and *Reasoner* ($\mathcal{R}$) offers theoretical advantages over monolithic distillation in terms of sample complexity and robustness against shortcut learning.

### 3.3.1 REDUCTION OF SAMPLE COMPLEXITY VIA DECOMPOSITION

Let $\mathcal{C}(f, \epsilon)$ denote the sample complexity required to learn a target function $f$ within an error bound $\epsilon$. In a monolithic setting, the student attempts to learn the complex mapping $f_{\text{mono}} : \mathcal{Q} \to \mathcal{R}$, which implicitly encompasses decomposition, retrieval, and reasoning. High-dimensional mappings with such composite internal structures often exhibit high Lipschitz constants or high intrinsic dimensions, requiring exponentially more data to generalize (Poggio et al., 2017).

In our modular framework, we approximate the target mapping as a composition of three simpler functions:

$$f_{\text{comp}}(q) = f_{\mathcal{R}}\left(q, f_{\mathcal{K}}(f_{\mathcal{A}}(q)), f_{\mathcal{A}}(q)\right) \tag{9}$$

We propose that the subfunctions—$f_{\mathcal{A}}$ (query decomposition), $f_{\mathcal{K}}$ (knowledge retrieval), and $f_{\mathcal{R}}$ (reasoning given context)—lie on a structurally simpler method than the direct mapping $f_{\text{mono}}$.

**Proposition 1 (Decomposition Benefit)** *Assuming the complexities of the sub-tasks are additive rather than multiplicative with respect to the difficulty of the monolithic task, and that intermediate supervision is available, the sample complexity satisfies:*

$$\mathcal{C}(f_{\mathcal{A}}, \epsilon) + \mathcal{C}(f_{\mathcal{K}}, \epsilon) + \mathcal{C}(f_{\mathcal{R}}, \epsilon) \ll \mathcal{C}(f_{mono}, \epsilon) \tag{10}$$

This inequality suggests that for a fixed budget of teacher-generated data, the modular student is expected to achieve lower generalization error. By explicitly supervising the intermediate latent variables $s$ and $k$, we effectively reduce the hypothesis space for each module, preventing the student from searching for solutions that are inconsistent with the logical structure of the problem.

### 3.3.2 SHORTCUT LEARNING VIA CAUSAL REGULARIZATION

A well-known problem in training large language models is *shortcut learning*, where the model learns spurious correlations between the question $q$ and the reasoning trace $r$ without understanding the underlying logic. Formally, a monolithic model maximizes $P(r|q)$. If the training set contains artifacts where specific phrasings in $q$ statistically predict tokens in $r$, the model may ignore the causal reasoning path.

Our modular framework imposes a *structural regularization* on the learning process. By enforcing the computational graph $q \to s \to k \to r$, we explicitly model the causal mechanism of reasoning.

*Information Bottleneck:* By training the modules independently (Eqs. 5-7), we enforce that the Reasoner $\mathcal{R}$ must utilize the information contained in the subquestions $\{s\}$ and knowledge $\{k\}$ to minimize its loss. Even though $\mathcal{R}$ is conditioned on $q$, the auxiliary supervision on $\mathcal{A}$ and $\mathcal{K}$ ensures that the intermediate context $\{s, k\}$ contains high-fidelity semantic signals. This reduces the mutual information between the input $q$ and the output $r$ that is mediated purely by spurious shortcuts:

$$I_{\text{spurious}}(r; q) \leq I_{\text{monolithic}}(r; q) \tag{11}$$

This structural constraint forces the student to "show its work" not just in the output space, but also in the internal functional space, leading to improved interpretability and robustness particularly in out-of-distribution (OOD) scenarios where spurious correlations often fail.

## 4 EXPERIMENTS

### DATASETS

We evaluate our system across three benchmark datasets: OBQA (OpenBookQA), StrategyQA, and MedQA-USMLE. Each dataset poses different reasoning challenges, allowing us to test the per-

formance and robustness of our modular framework. Table 4 summarizes the key statistics and characteristics of the three datasets, distinguishing them by format, complexity, and reasoning requirements.

OpenBookQA is a multiple-choice question answering dataset focused on elementary science. Each question typically requires reasoning over a core science fact (the "open book") combined with external common-sense knowledge. The dataset emphasizes fact recall and simple inference, making it suitable for evaluating subquestion decomposition and targeted knowledge injection (Mihaylov et al., 2018).

StrategyQA consists of binary (yes/no) questions that require multi-hop and implicit reasoning. This dataset is particularly well-suited to evaluating the reasoning capabilities of the *Reasoner* module under uncertainty and incomplete evidence (Geva et al., 2021).

MedQA (USMLE) is a challenging multiple-choice question dataset sourced from the United States Medical Licensing Examination. The questions demand advanced medical reasoning and rely heavily on both the recall of factual information and clinical decision-making. We found this dataset especially demanding for the *Informant* module, because it's very detailed and specific to the medical field (Jin et al., 2021).

Statistics about the dataset is given Appendix A.2

EXPERIMENTS DETAILS

We evaluate two distillation frameworks under varying model sizes:

- **TMD$_X$ (Three-Module Distillation):** It refers to the proposed 3 module architecture with each module has $X$ billion parameters. The three modules work sequentially to perform the task pipeline.
- **DRD$_Y$ (Direct Reasoning Distillation):** A simpler baseline, where a single model with $Y$ billion parameters is fine-tuned end-to-end to directly output reasoning steps–no modular breakdown involved (Magister et al., 2022).
- **JRD$_Y$ (Joint Reasoning Distillation):** A monolithic baseline with Y parameters trained to generate the full sequence of subquestions, knowledge, and reasoning in a single pass. This setup utilizes the exact same training data as TMD but without modular separation, serving as a control to isolate the benefits of the modular architecture from data enrichment.

For example, TMD$_{3B}$ consists of three independently fine-tuned 3-billion-parameter models operating in sequence, whereas DRD$_{8B}$ uses a single 8-billion-parameter model performing direct reasoning. This setup allows us to analyze the trade-off between modularization and model size. For training, we use *Llama-3.2-1B-Instruct*[2] and *Llama-3.2-3B-Instruct*[3] to inspect the performance of our framework in different parameter sizes. To compare against a single-model baseline that performs direct reasoning via knowledge distillation (without modular decomposition), we selected *Llama-3.1-8B-Instruct*[4] as a fair reference point. This choice ensures comparable parameter scale, since our full pipeline includes three separately tuned models.

In addition to the main results, we report:

- **Upper bound**: Predictions directly generated by the teacher model (GPT-4 or DeepSeek-R1), assuming ideal performance.
- **Lower bound**: Outputs from the base models without any fine-tuning ie. zero-shot. It represents the raw output of the base LLMs as used in the modular (TMD) or end-to-end (DRD, JRD) settings.

**Fine-tuning details.** All modules were fine-tuned using LoRA (Low-Rank Adaptation) via the *peft*[5] library. The following configuration was used for each module:

---

[2]https://huggingface.co/meta-llama/Llama-3.2-1B-Instruct
[3]https://huggingface.co/meta-llama/Llama-3.2-3B-Instruct
[4]https://huggingface.co/meta-llama/Llama-3.1-8B-Instruct
[5]https://github.com/huggingface/peft

The models were fine-tuned with a batch size of 2 per GPU and a maximum sequence length of 512 tokens. We used the AdamW optimizer with a linear learning rate scheduler and set the learning rate to 1e-4. LoRA adaptation was applied with a rank of 16, $\alpha = 8$, and a dropout rate of 0.1. All models were trained for 3 epochs using mixed-precision (fp16) training. Gradient accumulation was used with 4 steps to simulate larger batch sizes, and a weight decay of 0.01 was applied to prevent overfitting. Our experiments were conducted on 4 NVIDIA RTX A4000 GPUs.

Input prompts were tokenized with padding to 512 tokens, and padding tokens were masked in the label space to avoid loss contribution.

## 4.1 EXPERIMENT RESULTS

We structure our empirical investigation around four guiding questions, each illuminating a distinct strength of the Three-Module Distillation (TMD) framework. Results are drawn from both fine-tuning and zero-shot evaluations (Table 1 and Table 2).

We structure our empirical investigation to test the hypothesis that architectural decomposition is the key to unlocking reasoning in small language models. Beyond standard performance comparisons, our experiments are designed to isolate specific variables: the intrinsic reasoning capability in zero-shot settings, the scalability of gains under supervision, robustness across different teacher paradigms (GPT-4o and DeepSeek-R1), and the critical distinction between data-driven versus architecture-driven improvements. By controlling for model size and training data across our baselines (DRD and JRD), we aim to demonstrate that the TMD pipeline offers a superior inductive bias for low resource environments.

| Model | GPT-4o | | | DeepSeek-R1 | | |
|---|---|---|---|---|---|---|
| | OBQA | StrategyQA | MedQA | OBQA | StrategyQA | MedQA |
| *Fine-Tuned Models* | | | | | | |
| $DRD_{1B}$ | $65.22_{\pm.37}$ | $62.33_{\pm.71}$ | $33.22_{\pm.31}$ | $63.39_{\pm.25}$ | $61.79_{\pm.81}$ | $33.02_{\pm.28}$ |
| $DRD_{3B}$ | $72.53_{\pm.27}$ | $64.12_{\pm.81}$ | $43.60_{\pm.44}$ | $70.78_{\pm.21}$ | $63.88_{\pm.74}$ | $42.27_{\pm.30}$ |
| $JRD_{3B}$ | $73.47_{\pm.13}$ | $64.88_{\pm.41}$ | $43.78_{\pm.22}$ | $71.32_{\pm.18}$ | $64.11_{\pm.58}$ | $42.78_{\pm.25}$ |
| $DRD_{8B}$ | $79.32_{\pm.54}$ | $68.94_{\pm.92}$ | $50.56_{\pm.34}$ | $78.67_{\pm.43}$ | $67.32_{\pm.72}$ | $47.84_{\pm.46}$ |
| $JRD_{8B}$ | $81.26_{\pm.50}$ | $70.94_{\pm.73}$ | $52.04_{\pm.41}$ | $80.35_{\pm.33}$ | $68.76_{\pm.82}$ | $50.54_{\pm.43}$ |
| $TMD_{1B}$ | $74.14_{\pm.67}$ | $64.34_{\pm1.12}$ | $44.84_{\pm.53}$ | $72.43_{\pm.42}$ | $63.12_{\pm.92}$ | $43.12_{\pm.52}$ |
| $TMD_{3B}$ | $\mathbf{82.20}_{\pm.38}$ | $70.48_{\pm1.02}$ | $\mathbf{53.23}_{\pm.44}$ | $\mathbf{81.18}_{\pm.44}$ | $70.03_{\pm.88}$ | $\mathbf{51.44}_{\pm.37}$ |
| *Teacher (Upper Bound)* | | | | | | |
| GPT-4o | 92.12 | 78.47 | 74.19 | – | – | – |
| DeepSeek-R1 | – | – | – | 90.28 | 76.14 | 72.88 |

Table 1: Accuracy (%) of fine-tuned distilled models and teacher upper bounds under **GPT-4o** and **DeepSeek-R1** supervision across OBQA, StrategyQA, and MedQA. We report mean ± std over 3 runs.

| Model | OBQA | StrategyQA | MedQA |
|---|---|---|---|
| *Zero-Shot Performance (Lower Bound)* | | | |
| $DRD_{1B}$ | $36.54_{\pm.38}$ | $47.23_{\pm.90}$ | $31.98_{\pm.24}$ |
| $DRD_{3B}$ | $53.88_{\pm.46}$ | $51.72_{\pm1.08}$ | $33.28_{\pm.53}$ |
| $DRD_{8B}$ | $64.17_{\pm.44}$ | $60.63_{\pm1.18}$ | $39.11_{\pm.67}$ |
| $TMD_{1B}$ | $56.45_{\pm.63}$ | $54.16_{\pm.92}$ | $35.57_{\pm.41}$ |
| $TMD_{3B}$ | $\mathbf{66.20}_{\pm.83}$ | $61.28_{\pm.74}$ | $\mathbf{42.11}_{\pm.54}$ |

Table 2: Zero-shot accuracy (%) of base models used in DRD, JRD and TMD architectures, before any fine-tuning. These represent lower bounds in our distillation setup. We report mean ± std over 3 runs.

We structure our empirical investigation around five guiding questions, each illuminating a distinct strength of the Three-Module Distillation (TMD) framework. Results are drawn from both fine-tuning and zero-shot evaluations (Table 1 and Table 2).

Q1: DOES MODULAR DISTILLATION IMPROVE ZERO-SHOT REASONING?

Zero-shot experiments demonstrate that TMD models consistently outperform both the DRD and the JRD baselines, suggesting that our proposed architecture improves reasoning ability internally. On the OBQA benchmark, $TMD_{3B}$ attains an accuracy of 66.20%, which notably surpasses both $DRD_{8B}$ at 64.17% and $JRD_{8B}$ at 64.89%.

It is particularly revealing that while JRD shows slight improvements over DRD due to its exposure to structured data–such as $JRD_{3B}$ scoring 55.20% on OBQA compared to 53.88% for $DRD_{3B}$–it still falls significantly short of the performance achieved by $TMD_{3B}$. Furthermore, even the smaller $TMD_{1B}$ achieves 54.16% on StrategyQA, competing closely with the much larger $JRD_{3B}$, which scores 54.72%. These results indicate that the modular separation of Analyzer, Informant, and Reasoner allows smaller models to solve complex problems more effectively than simply training a monolithic model on the same data.

Q2: DOES FINE-TUNING AMPLIFY MODULAR ADVANTAGES?

Supervised fine-tuning under *Teacher* instruction leads to substantial gains across all models, yet TMD continues to maintain a clear lead over both DRD and JRD baselines. Under GPT-4o supervision, $TMD_{3B}$ reaches a score of 82.20% on OBQA, effectively outperforming the significantly larger $JRD_{8B}$ and $DRD_{8B}$, which score 81.26% and 79.32%, respectively.

We observe a similar trend under DeepSeek-R1 supervision. $TMD_{3B}$ achieves 81.18% on OBQA and 70.03% on StrategyQA. In contrast, the $JRD_{8B}$ baseline, despite having access to the exact same intermediate reasoning data, lags behind with scores of 80.35% on OBQA and 68.76% on StrategyQA. This highlights a critical finding: simply feeding intermediate reasoning steps to a single model faces diminishing returns due to capacity bottlenecks, while modularizing these steps unlocks higher performance.

Q3: IS TMD ROBUST ACROSS DIFFERENT SUPERVISION SOURCES?

To evaluate TMD's performance across different datasets, we use two distinct teacher models: GPT-4o and DeepSeek-R1. The results show clear trend: TMD performs robustly under the supervision of both teachers, consistently surpassing both baselines.

Under GPT-4o supervision, $TMD_{3B}$ surpasses the monolithic $JRD_{8B}$ model across all three datasets, achieving 82.20% on OBQA and 53.23% on MedQA, compared to 81.26% and 52.04% for the baseline. When fine-tuned with DeepSeek-R1, $TMD_{3B}$ maintains this superiority, scoring 51.44% on MedQA versus 50.54% for $JRD_{8B}$. Although DeepSeek-R1 generally yields slightly lower performance compared to GPT-4o across all students, the relative performance ranking remains unchanged, demonstrating that the architectural advantage of TMD is robust to the choice of the teacher model.

Q4: HOW DO FLOPS AND MEMORY USAGE INFLUENCE THE ADVANTAGE OF TMD?

Figure 2 illustrates the analysis of performance relative to computational resources. The data clearly demonstrates the efficiency of our modular framework. While $JRD_{3B}$ and $DRD_{3B}$ operate within a similar computational budget, TMD consistently delivers significantly higher accuracy.

Crucially, although TMD utilizes three distinct modules, its sequential execution allows it to achieve superior performance metrics without the resource penalties typically associated with larger models. $TMD_{3B}$ achieves accuracy levels that surpass those of $JRD_{8B}$ and $DRD_{8B}$, effectively delivering large-model performance with a small-model footprint. This characteristic makes TMD particularly attractive for deployment in real-world settings where maximizing performance within strict hardware constraints is essential.

Q5: IS THE PERFORMANCE GAIN DUE TO DATA ENRICHMENT OR MODULAR ARCHITECTURE?

A key question is whether TMD's success stems from the modular architecture itself or simply from the richer training data distilled from the teacher. The JRD baseline serves as the control experiment

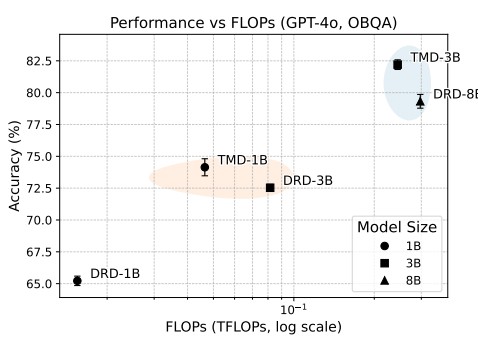 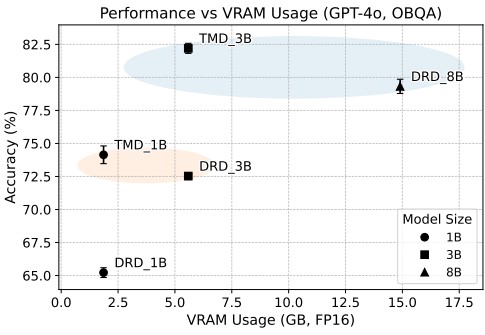

(a) Performance vs FLOPs (GPT-4o, OBQA).     (b) Performance vs VRAM usage (GPT-4o, OBQA).

Figure 2: Comparison of distilled models under GPT-4o supervision on OBQA. (a) Accuracy as a function of inference FLOPs. (b) Accuracy as a function of estimated VRAM usage. Ellipses highlight the key comparison pairs (TMD vs DRD).

for this, as it is trained on the exact same enriched data sequence as TMD but within a monolithic architecture.

Comparing $TMD_{3B}$ and $JRD_{3B}$ reveals a substantial performance gap. On the MedQA benchmark supervised by GPT-4o, $TMD_{3B}$ scores 53.23%, whereas $JRD_{3B}$ only reaches 43.78%–a difference of nearly 10 percentage points. Similarly, on OBQA, TMD leads with 82.20% while JRD trails at 73.47%.

This confirms that data enrichment alone is insufficient for small models. A single 3B model suffers from significant task interference when trying to simultaneously learn decomposition, retrieval, and reasoning tasks. By decoupling these tasks into specialized modules, TMD mitigates this interference, allowing each component to master its specific domain. Thus, the modular architecture is the primary driver of the observed performance gains.

## 4.2 ABLATION STUDY

To better understand how each of the three modules contribute to the overall system, we ran an ablation study. We tested different combinations where some modules were fine-tuned and others were used zero-shot (without fine-tuning). We did this on three different QA datasets: OBQA, StrategyQA, and MedQA. The results are shown in Table 3 for two teacher models: GPT-4o and DeepSeek-R1.

In the table, a checkmark (✓) means that the module was fine-tuned, while a cross (✗) means the zero-shot module was used. All models have 3 billion parameters ($TMD_{3B}$ architecture).

| *Analyzer* | *Informant* | *Reasoner* | **GPT-4o** | | | **DeepSeek-R1** | | |
|:---:|:---:|:---:|:---:|:---:|:---:|:---:|:---:|:---:|
| | | | OBQA | StrategyQA | MedQA | OBQA | StrategyQA | MedQA |
| ✓ | ✗ | ✗ | 69.12 | 64.61 | 43.41 | 67.58 | 64.10 | 43.11 |
| ✗ | ✓ | ✗ | 73.14 | 64.91 | 45.89 | 72.63 | 63.58 | 45.42 |
| ✗ | ✗ | ✓ | 71.16 | 66.80 | 45.51 | 70.98 | 66.36 | 45.38 |
| ✓ | ✓ | ✗ | 76.90 | 65.74 | 48.21 | 76.56 | 65.82 | 47.88 |
| ✓ | ✗ | ✓ | 76.53 | 69.44 | 46.72 | 76.12 | 68.80 | 46.03 |
| ✗ | ✓ | ✓ | 78.44 | 69.24 | 51.16 | 77.17 | 68.71 | 50.74 |

Table 3: TMD ablation study on OBQA, StrategyQA, and MedQA using **GPT-4o** and **DeepSeek-R1** teachers. ✓ indicates the corresponding module is fine-tuned; ✗ indicates the corresponding module is zero-shot. All models are 3B.

**Single-Module Contributions.** Fine-tuning each module separately improves performance over the zero-shot baseline, although the size of the gain depends on the dataset. The *Analyzer* gives important improvements on OBQA and StrategyQA. Its effect on MedQA is smaller, since analyzing

the context alone is not enough for medical reasoning. The *Informant* consistently outperforms the *Analyzer* across all tasks, and is particularly effective on MedQA, where it reaches 45.89%, shows the importance of domain knowledge for datasets. The *Reasoner* shows the strongest single-module performance, especially on StrategyQA with a score of 66.80%, demonstrating its ability to handle complex reasoning even without external knowledge.

**Two-Module Combination.** Using two modules together generally gets better results than relying on a single module, though the most effective pairing varies by dataset. The *Analyzer + Informant* combination performs well on OBQA, achieving 76.90%. It is expected due to its strength in organizing and contextualizing factual information. However, this pair is less effective on StrategyQA and MedQA datasets, where deeper reasoning is required to get final answer. The *Analyzer + Reasoner* pair performs the best results on StrategyQA with 69.44%, suggesting that the synergy between question decomposition and reasoning is especially beneficial for abstract or implicit questions. On MedQA, the highest-performing two-module setup is *Informant + Reasoner*, which achieves 51.16%. This highlights the value of combining domain-specific knowledge with reasoning capabilities when addressing complex medical questions.

**Dataset-Specific Module Importance.** The ablation results reveal a clear pattern: the contribution of each module shifts depending on the nature of the task. This confirms that our framework adapts to the specific "bottleneck" of each dataset.

For open-domain tasks like OBQA, the primary challenge is retrieving the right information. Consequently, performance relies heavily on the *Analyzer* to decompose the query and the *Informant* to fetch relevant facts. In contrast, StrategyQA is a test of implicit logic rather than simple fact-checking. Here, the *Reasoner* becomes the dominant driver, as the model must chain multiple logical steps together to reach a conclusion.

MedQA presents the most complex scenario because it demands the *Informant* to recall precise medical information and the *Reasoner* to apply clinical judgment. Removing either module leads to a sharp performance drop, illustrating that specialized domains require both high-fidelity knowledge and robust logic.

## 5 CONCLUSION

In this work, we introduced a modular distillation framework that bridges the gap between the reasoning capabilities of large teacher models and efficient student models. By decomposing the monolithic reasoning process into three specialized components, we demonstrated that small language models can achieve robust performance on knowledge-intensive tasks without requiring massive computational costs. Our empirical results across OBQA, StrategyQA, and MedQA confirm that structural decomposition is a powerful alternative. Beyond quantitative gains, our framework offers a significant advantage. Unlike black box monolithic models, our approach exposes the logical flow of decomposition, retrieval, and synthesis, making the reasoning process transparent and verifiable.

Our results suggest that small models are limited less by capacity than by a lack of structural guidance. By providing a new reasoning architecture, we unlock their potential to navigate complex tasks. We hope this encourages a move toward modular designs, leading to AI that is efficient, flexible, and transparent.

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

# A APPENDIX

## A.1 PROMPT TEMPLATES

This appendix provides the prompt templates used to guide each of the three modules in our distillation framework: *Analyzer*, *Informant*, and *Reasoner*. Each prompt is carefully designed to elicit module-specific behavior while maintaining consistency across the overall reasoning pipeline.

Analyzer Prompt The *Analyzer* module is responsible for decomposing complex questions into focused subquestions. This decomposition facilitates targeted knowledge retrieval and modular reasoning. The prompt encourages the generation of only relevant subquestions, with a fallback response when decomposition is unnecessary.

```
Your task is to break down a given complex question
into the most relevant and helpful subquestions.
You will also consider the provided options to
generate subquestions that aid in understanding
and solving the main question effectively. Only
return subquestions that directly aid in answering
the original question, avoiding any that could be
harmful or irrelevant. If the question does not need
breaking down to be answered, return 'No decomposition'.
Otherwise, strictly list the necessary subquestions.
Question: {question}
Options: {options}
Write Subquestions
```

### A.1.1 INFORMANT PROMPT

The *Informant* module generates concise, focused knowledge snippets in response to each subquestion. Its goal is to surface grounded and relevant background information without unnecessary elaboration. The prompt emphasizes brevity, precision, and the avoidance of speculation.

```
You are an expert assistant with a vast knowledge base.
For the given question, provide a short, concise, and
relevant background without adding any extra
information or questions.
Question: {subquestion}
Write Relevant Knowledge
```

### A.1.2 REASONER PROMPT

The *Reasoner* module performs structured reasoning by integrating the main question, candidate options, subquestions, and retrieved knowledge. The prompt enforces a format that encourages clarity, step-by-step inference, and selection of a final answer from the given options. This supports interpretability and traceability of the reasoning process.

```
You are an expert assistant specializing in reasoning
and providing structured answers. Given a main question,
options, subquestions, and relevant knowledge, determine
the correct option based on the reasoning process.
Strictly adhere to the provided format.
Provide the final answer as one of the given
options (e.g., 'ending0', 'ending1').
Keep the reasoning concise and structured.
Main Question: {question}
Options: {endings}
Subquestions and Relevant Knowledge:
{subquestions and knowledge}
Write Reasoning and Final Answer
```

### A.1.3 DIRECT REASONING DISTILLATION PROMPT

The *Reasoner* module performs structured reasoning by integrating the main question, candidate options, subquestions, and retrieved knowledge. The prompt enforces a format that encourages clarity, step-by-step inference, and selection of a final answer from the given options. This supports interpretability and traceability of the reasoning process.

```
You are given a multiple-choice question and possible
```

```
answer options. Your task is to reason through the
question in a clear, structured way using numbered steps.
Each step should be factual, concise, and contribute
to evaluating the correctness of the options.
The reasoning should resemble a scientific or
biological explanation if relevant.
After the numbered reasoning,
conclude with the Final Answer using the format:

Final Answer: [Correct Option Letter]

Question: {question}
Options: {options}

Write Reasoning and Final Answer
```

## A.2 DATASET STATISTICS

| Dataset | # Examples | Format | Reasoning Type |
| --- | --- | --- | --- |
| OBQA | 4957 train / 500 val / 500 test | 4-way MCQ | 2–3-step inference using core science facts and common-sense knowledge |
| StrategyQA | 1603 train / 687 val / 687 test | Binary (Y/N) | Implicit multi-hop reasoning requiring strategic decomposition |
| MedQA | 10178 train / 1272 val / 1273 test | 4-way MCQ | Expert-level multi-step clinical reasoning |

Table 4: Overview of datasets used in experiments.