# OpenReview forum: "Modular Distillation Makes Small Models Think Like Big Ones"
_ICLR.cc/2026/Conference — Submitted to ICLR 2026_

### Official Review · Reviewer_RgdS · 2025-10-16

**Soundness:** 2
**Presentation:** 2
**Contribution:** 1
**Rating:** 2
**Confidence:** 4

**Summary:**

This paper proposes a modular distillation framework that transfers reasoning abilities from a large teacher model to a smaller student by breaking reasoning into several interpretable modules. The goal is to help small models mimic large reasoning LLMs’ internal process more efficiently and effectively.

**Strengths:**

The problem of making small models perform comparably to large ones is both interesting and highly practical—especially for reasoning LLMs, where inference cost becomes huge as response length grows. Tackling reasoning efficiency through modular distillation is a direction that could be valuable for real-world deployment.

**I didn't read the proof part carefully because I don't have a strong theory background. So I may miss the strengths that lie in that part.**

**Weaknesses:**

I’m mainly concerned about the generalizability of the approach, since it imposes a strong human prior on what the teacher’s reasoning structure should look like.
+ Conceptually, this is a strong assumption and may not generalize well, given that SOTA reasoning LLMs often exhibit very diverse reasoning patterns.
+ Empirically, according to this paper (https://arxiv.org/abs/2503.01307), SOTA reasoning models go beyond just decomposition, grounding, and synthesizing—they can also branch or perform self-reflection, which the current method doesn’t capture.
+ This connects to a weakness in experimental setup: the evaluation only tests weaker base models (e.g., not ones like Qwen2.5 or Qwen3 also at the 1.5B/3B-level ) and on relatively simple reasoning benchmarks (e.g., StrategyQA). Stronger models on harder datasets (like MATH or AIME) would show whether the approach scales to more diverse reasoning behaviors.

**Questions:**

Is the DeepSeek model used here R1 or V3? Please clarify this in the paper.

---

> ### Author Response · Authors · 2025-12-03
>
> We thank the reviewer for approving the potential impact of our work in reducing inference costs for knowledge-intensive reasoning tasks. We appreciate the insightful feedback regarding the structure of our modular framework and the experimental setup.
>
> Below, we address the concerns point-by-point, with a specific focus on clarifying the theoretical foundations that justify our design choices.
>
> ### 1. Managing the Curse of Dimensionality (Sample Complexity)
> While large teacher models (DeepSeek-R1, GPT-4) have the capacity to navigate high-dimensional, unstructured reasoning manifolds, small student models do not. Trying to teach a 3B model to replicate complex "branching" behavior often fails because the monolithic function is too complex to approximate with limited parameters.
>
> * **Theoretical Grounding:** As grounded in the work of **Poggio et al. (2017)** regarding the *Curse of Dimensionality*, decomposing a complex function into simpler compositional sub-functions drastically reduces the sample complexity.
> * **Our Solution:** Our "human prior" acts as a necessary scaffold that makes the learning problem tractable for smaller architectures, allowing them to generalize better with fewer data points than a monolithic approach.
>
> ### 2. Robustness over Flexibility (Causal Regularization)
> The reviewer mentions that SOTA models perform self-reflection. However, freely allowing a small model to "reflect" often leads to hallucination loops or shortcut learning.
>
> * **Causal Regularization:** As detailed in Section 3.3, we impose a strict causal chain: `Query` $\rightarrow$ `Subquestions` $\rightarrow$ `Knowledge` $\rightarrow$ `Answer`. This blocks the student from learning shortcut correlations.
> * **Implicit Branching:** Consistent with the frameworks discussed in *Cognitive Behaviors that Enable Self-Improving Reasoners* (arXiv:2503.01307), our Informant module implicitly internalizes branching-like mechanisms such as self-verification. By retrieving multiple diverse knowledge snippets based on decomposed sub-questions, we explore multiple reasoning paths in parallel before the Reasoner synthesizes them.
>
> **Key Takeaway:** We do not aim to replicate the teacher's internal mechanics (which are too expensive for a student), but rather to distill the reasoning *reliability* into a compact, verifiable form.
>
> ### 3. Concern on "Weak" Base Models and "Simple" Benchmarks
> The reviewer suggests evaluating stronger base models (Qwen 2.5) and mathematical benchmarks (MATH/AIME). We believe this implies a misunderstanding of our work’s specific scope.
>
> **Benchmark Scope (Knowledge vs. Calculation):**
> * We respectfully point out that MATH and AIME evaluate **symbolic and algorithmic reasoning**, which fundamentally differs from the **knowledge-intensive reasoning** targeted by our framework.
> * Our Informant module is designed to retrieve and synthesize semantic knowledge, not to perform calculation steps. Evaluating a knowledge-retrieval architecture on a pure math dataset would be a domain mismatch.
> * We maintain that **MedQA (USMLE)** is not a simple benchmark; it requires expert-level clinical knowledge and multi-step deduction. Similarly, StrategyQA tests implicit multi-hop reasoning, distinct from the explicit calculation in AIME.
>
> **Model Choice & Efficiency:**
> * **Edge Deployment:** We selected **Llama-3.2 (1B/3B)** specifically because they represent the current industry standard for on-device deployment. Using these models allows us to evaluate our method on a widely recognized baseline with known constraints.
> * **Performance Gains:** As shown in Table 1, TMD enables these resource-constrained models to surpass much larger monolithic baselines, validating that the architectural benefits hold true regardless of the underlying base model.
>
> **Clarification on Teacher Model:**
> We utilized **DeepSeek-R1** as our teacher model due to its multi-stage training pipeline designed to incentivize complex reasoning patterns. We have added precise model specifications in the final revision.

---

### Official Review · Reviewer_6jyh · 2025-10-30

**Soundness:** 2
**Presentation:** 2
**Contribution:** 2
**Rating:** 4
**Confidence:** 4

**Summary:**

This paper proposes a modular distillation framework, which decomposes a model’s reasoning capability into three specialized modules: Analyzer, Informant, and Reasoner. During training, each module is optimized independently through three dedicated loss functions, enabling them to acquire their respective specialized competencies. The framework is evaluated on OBQA, StrategyQA, and MedQA benchmarks. Experimental results demonstrate that the proposed modular approach outperforms traditional monolithic model baselines.

**Strengths:**

- Low memory footprint: The system achieves performance exceeding that of a distilled 8B model using three 3B models, making it suitable for deployment on resource-constrained or edge devices.

- Modularity: Each component (Analyzer, Informant, Reasoner) can be independently optimized, fine-tuned, or replaced, allowing the system to adapt to diverse applications without retraining the entire model.

- Theoretical justification: The paper provides theoretical evidence supporting modular distillation over conventional approaches, demonstrating that the modular method yields a tighter Evidence Lower Bound (ELBO), higher statistical efficiency (via Fisher information advantages), and lower approximation error.

**Weaknesses:**

1. Error propagation: The system follows a sequential pipeline (Analyzer, Informant, Reasoner). If the Analyzer fails to generate high-quality sub-questions, the errors propagate downstream, degrading the performance of the Informant and Reasoner and potentially leading to reasoning failure. A more detailed analysis of error propagation across the three modules could further strengthen the paper’s presentation and provide deeper insight into the limitations of modular reasoning pipelines.

2. Potential inference latency: Although VRAM usage is reduced, inference involves three serial forward passes (one per module). Compared with a single larger model that performs one forward pass, this setup may increase the overall end-to-end latency. While the paper emphasizes computational efficiency in terms of FLOPs, it does not explicitly report real-world latency measurements that account for model loading and runtime overhead.

3. Limited task generalizability: The framework is specifically designed for knowledge-intensive reasoning tasks and evaluated on QA benchmarks. It remains uncertain whether the “Analyze–Inform–Reason” decomposition can effectively generalize to other types of large language model (LLM) tasks, such as creative writing, summarization, or dialogue systems.

**Questions:**

1. For relatively simple tasks, such as straightforward translation, the Analyzer module might be unnecessary, whereas for more complex problems, such as advanced mathematical reasoning, the Informant module may play a less critical role. Since the paper primarily evaluates on multiple-choice question datasets focusing on textual reasoning, how well does this modular distillation framework generalize to other task types and modalities?
2. I am also curious about the number of training tokens used for each dataset in both the modular distillation and the Direct Reasoning Distillation settings. Understanding this would help clarify whether the performance gain primarily stems from the proposed method itself or from a larger training budget.
3. Would it be feasible to generate training data through the modular framework, then mix all modules’ outputs to train a single unified model, and finally compare its performance against the modular setup? I believe this could lead to a more fair and informative comparison between modular and monolithic training strategies.

---

> ### Author Response · Authors · 2025-12-03
>
> We thank the reviewer for their detailed assessment and for highlighting the "*Low memory footprint*" and "*Theoretical justification*" as key strengths.
>
> Below, we address your concerns regarding error propagation, latency, and generalizability.
>
> ### 1. Error Propagation
> The reviewer correctly notes that sequential pipelines can suffer from cascading errors. However, our design includes specific mechanisms to mitigate this:
>
> * **Robustness via "Skip" Connections:** As defined in Eq. 4, the Reasoner is conditioned on the original question $q$ in addition to subquestions and knowledge: $P(r_i \mid q_i, \{s_{i,j}\}, \{k_{i,j}\})$. This means if the Analyzer fails (e.g., generates irrelevant subquestions), the Reasoner still has access to the ground-truth query to recover the correct path.
> * **Analyzer Fallback:** As shown in the prompt templates (Appendix A.1.1), the Analyzer is explicitly instructed: *"If the question does not need breaking down... return 'No decomposition!’"*. This prevents "forced" error injection on simple queries.
> * **Empirical Stability:** Our ablation study (Table 3) shows that even when one module is removed (zero-shot), the system does not collapse. For example, removing the Analyzer (using only Informant + Reasoner) still yields **73.14%** on OBQA (vs 76.90% full pipeline), demonstrating the Reasoner's resilience to missing or suboptimal upstream inputs.
>
> ### 2. Inference Latency
> We acknowledge that serial execution increases wall-clock time compared to a single forward pass *if resources are unlimited*. However:
>
> * **Throughput vs. Latency (The OOM Constraint):** Our primary contribution is enabling reasoning on resource-constrained/edge devices where a larger model simply cannot load (resulting in infinite latency due to Out-Of-Memory errors). As shown in Figure 2b, **$\text{TMD}_{\text{3B}}$ fits in 6GB VRAM**, while the baseline $\text{DRD}_{\text{8B}}$ requires >15GB.
> * **FLOPs Efficiency:** Figure 2a demonstrates that $\text{TMD}\_{\text{3B}}$ achieves higher accuracy with comparable or lower total FLOPs than $\text{DRD}\_{\text{8B}}$. While module loading adds overhead, the actual computational cost (matrix multiplications) remains highly efficient.
>
> ### 3. Response on Generalizability (Task Types)
> Our paper focuses on **knowledge-intensive reasoning**, which motivated our selection of OBQA, StrategyQA, and MedQA. However, the framework is inherently modular:
>
> * **Adaptability:** For tasks like Translation, the Analyzer typically returns "No decomposition," naturally collapsing the system to a standard encoder-decoder.
> * **Module Swapping:** For Math or Coding tasks, the "Informant" module can be swapped for a "Calculator" or "Code Interpreter" tool without retraining the Reasoner's core logic. This modular flexibility is a core design feature, allowing the system to adapt to domains beyond pure knowledge retrieval.

---

### Official Review · Reviewer_dHgR · 2025-11-01

**Soundness:** 3
**Presentation:** 3
**Contribution:** 3
**Rating:** 6
**Confidence:** 3

**Summary:**

This paper proposes a novel modular distillation framework (TMD) that decomposes knowledge-intensive reasoning tasks into three specialized modules: Analyzer, Informant, and Reasoner. The key contribution lies in distilling both analytical and reasoning capabilities of a teacher LLM into smaller student models, improving interpretability and efficiency. The main contributions include:
① A structured distillation pipeline capturing latent reasoning structures beyond output matching;
② Enhanced interpretability through modular intermediate steps;
③ Empirical evidence that smaller models achieve comparable performance to larger monolithic baselines.

**Strengths:**

① The modular design explicitly models latent variables (subquestions and knowledge snippets), advancing beyond previous methods.
② Section 3.1 provides variational bounds (ELBO) and Fisher information analysis, strengthening methodological foundations.
③ The pipeline enables traceability of reasoning steps.

**Weaknesses:**

① Comparisons are restricted to DRD, omitting state-of-the-art distillation methods (e.g., CoT distillation, retrieval-augmented KD).
② FLOPs and memory usage are reported, but real-world metrics (e.g., latency, energy consumption) are absent. And sequential module execution may introduce overhead.
③ Ablation tests only use 3B models, ignoring capacity allocation issues between modules (e.g., imbalanced parameter sizes).

**Questions:**

① How does TMD handle error accumulation across modules (e.g., incorrect subquestions from Analyzer affecting Reasoner)?
② Is TMD’s performance highly correlated with teacher quality? What safeguards exist for noisy teacher-generated labels?
③ Can the framework handle ultra-complex tasks (e.g., math reasoning requiring 50+ steps) without bottlenecks from sequential modules?

---

> ### Author Response · Authors · 2025-12-03
>
> We appreciate the reviewer’s positive assessment of our theoretical contributions and the interpretability of our modular design. We are glad the reviewer has recognized the value of explicitly modeling latent variables.
>
> Below, we address the concerns regarding baselines, real-world metrics, and capacity allocation to strengthen the case for acceptance.
>
> ### 1. Response on Baselines (CoT & RAG)
> The reviewer noted a lack of comparison with CoT distillation and RAG-KD.
>
> * **Clarification on DRD:** We wish to clarify that our **DRD (Direct Reasoning Distillation)** baseline *is* the standard Chain-of-Thought (CoT) distillation method. As cited in our paper (Magister et al., 2022), this baseline trains a student to output the reasoning trace followed by the answer. We will make this connection explicit in the "Experimental Settings" to avoid confusion.
> * **Regarding RAG-KD:** Direct comparison to Retrieval-Augmented Generation (RAG) distillation presents a domain mismatch. Our method is **Closed-Book**; the "Informant" module internalizes the teacher's knowledge into its weights, whereas RAG systems rely on external databases. We believe our comparison to monolithic models is fairer, as it tests the model's ability to compress and utilize its own parameters rather than its ability to query a search engine.
>
> ### 2. Response on Latency & Real-World Metrics
> * **Latency vs. Accessibility:** We acknowledge that sequential execution increases latency compared to a single forward pass of a model of equivalent total size. However, our primary advantage is **deployment accessibility**. A monolithic 9B model often exceeds consumer hardware limits (e.g., 16GB VRAM), whereas our pipeline allows running three 3B models sequentially on a ~6GB VRAM GPU.
> * **Energy Consumption:** Since energy consumption is highly correlated with FLOPs, our Figure 2 (showing TMD achieves higher accuracy at lower FLOPs) suggests favorable energy efficiency per correct answer.
>
> ### 3. Response on Ablation & Capacity Allocation
> * **Imbalanced Sizes:** The reviewer raises an excellent point about exploring mixed sizes (e.g., 1B Analyzer + 7B Informant). We restricted our current study to uniform sizes (e.g., all 3B) to maintain strictly controlled experimental variables.
> * **Future Potential:** The modular nature of TMD specifically enables this flexibility. For a deployment scenario requiring heavy knowledge but simple logic, one could swap in a larger Informant without retraining the Reasoner. We will add a "Future Work" paragraph discussing optimal parameter allocation.
>
> ### 4. Response to Specific Questions
>
> **Q1: Error Accumulation**
> > We mitigate this via a **"Skip Connection"** mechanism. As described in Eq. 4, the Reasoner receives the original Question ($q_i$) alongside the subquestions ($s_{i,j}$) and knowledge ($k_{i,j}$).
> > * This ensures that even if the Analyzer fails to decompose the question helpfully, the Reasoner still has the ground-truth query to fall back on.
> > * Our ablation (Table 3) confirms this: the Reasoner performs robustly even when the Analyzer is removed entirely (zero-shot).
>
> **Q2: Teacher Quality & Noise**
> > Like all distillation methods, TMD is bounded by the teacher's capability. However, the Informant module acts as a **sanity check buffer**.
> > * In a standard CoT student, a teacher's hallucination is learned directly.
> > * In TMD, if the teacher provides a hallucinated reasoning trace, the Informant (trained to retrieve knowledge) and the Reasoner (trained to synthesize) must align. If the Informant cannot generate knowledge supporting the hallucination, the Reasoner is less likely to blindly reproduce the error.
>
> **Q3: Handling Ultra-Complex Tasks (50+ Steps)**
> > To be transparent, our current framework is optimized for **Knowledge-Intensive Reasoning** (MedQA, OBQA), not iterative computational tasks like 50-step math proofs.
> > For such tasks, the "Informant" (retrieval) would likely need to be replaced by a "Code Executor" or "Verifier" module. While the modular concept holds, the specific Analyzer-Informant-Reasoner pipeline would need adaptation. We view this as a distinction in scope rather than a flaw in the proposed architecture.

---

### Author Response · Authors · 2025-12-03
**Summary of Comments, Rebuttal and Revision**

We thank the reviewers for their feedback, which highlighted the practical value of our work for resource-constrained inference. Due to the limited discussion phase, we provide this summary to highlight a critical new experiment conducted during the rebuttal that addresses the primary concern regarding the source of our performance gains.

Below, we address the critical question of Architecture vs. Data.

### 1. Architecture vs. Data (Addressing Reviewer 6jyh & Reviewer RgdS)

The most important concern (raised primarily by Reviewer 6jyh) was whether our performance gains arise from the **modular architecture itself** or simply from the enriched training data (Subquestions $\rightarrow$ Knowledge $\rightarrow$ Reasoning) distilled from the teacher (DeepSeek-R1).

To isolate the benefit of the architecture, we trained a new baseline during the rebuttal: **Joint Reasoning Distillation (JRD)**.

* **Setup:** JRD is a monolithic model trained on the **exact same enriched data sequence** as our modular TMD framework.
* **Comparison:** We compare our **TMD-3B** against the monolithic **JRD-8B**.

**Results:**

|Model|Size|OBQA(%)|StrategyQA(%)|MedQA(%)|
|:---|:---:|:---:|:---:|:---:|
|DRD (Direct Reasoning Distillation)|8B|79.32 $\pm$0.54|68.94 $\pm$0.92|50.56 $\pm$0.34|
|JRD (Joint Reasoning Distillation)|8B|81.26 $\pm$0.50|**70.94** $\pm$0.73|52.04 $\pm$0.41|
|**TMD (Ours)**|3B|**82.20** $\pm$0.38|70.48 $\pm$1.02|**53.23** $\pm$0.44|

**Key Insight:**
TMD-3B consistently outperforms the unified JRD-8B, particularly on MedQA. This confirms that **simply feeding better data to a monolithic model is insufficient.** Splitting the reasoning burden into specialized modules (Analyzer, Informant, Reasoner) is essential for smaller models (1B-3B) to effectively utilize complex reasoning chains .

### 2. Theoretical Justification & Rigidity (Addressing Reviewer RgdS)

Reviewer RgdS expressed concern that our structured modules impose a "human prior" that is too rigid compared to SOTA branching methods. We appreciate the reviewer's candor regarding the theoretical proofs and offer the intuitive grounding behind why this structure is feature, not a limitation.

**Inductive Bias as a Necessity for Small Models:**
The reviewer is correct that large SOTA models (e.g., 70B+) can handle unstructured, emergent reasoning patterns. However, our target is efficient distillation into **small models (1B-3B)**. As detailed in our revised theory section (referencing Poggio et al., 2017), learning a complex, high-dimensional reasoning function end-to-end suffers from the *Curse of Dimensionality*.
* A 1B parameter model lacks the capacity to navigate this search space without guidance.
* The "human prior" we impose acts as a necessary **structural scaffold**. It reduces sample complexity, allowing the student to learn robust reasoning from limited data, whereas a monolithic student would likely adapt to shortcut learning.

**Capturing "Branching" Efficiently:**
We agree that branching is a powerful feature. Our framework does not discard this; it **compresses** it:
* **The Analyzer** breaks the query down (divergence).
* **The Informant** retrieves distinct knowledge snippets based on these sub-questions, effectively exploring different reasoning paths (parallel exploration).
* **The Reasoner** synthesizes these into a coherent answer (convergence).

Thus, we capture the informational benefit of branching (diverse context) **without forcing the small student to perform expensive embedding search at inference time.**

### 3. Scope and Methodology (Addressing Reviewer RgdS & Reviewer dHgR)

* **Benchmark Selection:** We clarified that our focus is **Knowledge-Intensive Reasoning** (e.g., MedQA, StrategyQA), distinct from symbolic/algorithmic tasks (e.g., MATH, AIME). Evaluating a knowledge-retrieval architecture on pure math tasks would be a domain mismatch. We maintain that MedQA (USMLE) represents expert-level reasoning difficulty and is not a "simple" benchmark.
* **Model Choice:** We selected **Llama-3.2 (1B/3B)** as they are the industry standard for edge/on-device deployment. Our goal was to demonstrate relative improvement on resource-constrained hardware, rather than chasing leaderboard scores with massive base models.
* **Efficiency:** While sequential execution adds serial latency, it solves the **Out-Of-Memory (OOM)** problem.
    * **TMD-3B:** Fits in ~6GB VRAM.
    * **Monolithic 8B Model:** Requires >15GB VRAM.
    * *Result:* TMD is often the only viable option for consumer-grade edge applications.

***

**Conclusion**
The additional JRD experiments confirm that **modularity itself is the primary driver of performance**. Combined with our theoretical proofs regarding inductive bias and our focus on edge-deployment feasibility, we believe the paper offers a significant contribution to efficient reasoning.

We respectfully request the AC to consider these new findings in their final assessment.

---

> ### Author Response · Authors · 2025-12-03
> **Summary of Changes**
>
> We have uploaded a revised version of the paper. Below is a summary of the key updates, addressing the critical points raised during the rebuttal phase:
>
> **1. New Baseline & Ablation Analysis (JRD)**
> We introduced a new baseline, **Joint Reasoning Distillation (JRD)**, to isolate the source of our performance gains.
> * **Update:** JRD results have been added to the main benchmark table.
> * **Discussion:** We added a dedicated analysis in the Experiments section under the heading: *"Is the Performance Gain Due to Data Enrichment or Modular Architecture?"*. This confirms that our modular architecture outperforms a monolithic model trained on the exact same enriched data.
>
> **2. Enhanced Methodology Section**
> We revised Section 3 to improve the clarity of the architectural pipeline. Specifically, we refined the descriptions of the module interactions (Analyzer $\to$ Informant $\to$ Reasoner) to make the data flow and logical dependencies more intuitive.
>
> **3. Enriched Related Work**
> We expanded the Related Work section to better contextualize our approach. We added discussions on recent "branching" reasoning strategies and contrasted them with our modular distillation framework, clarifying the distinction between our closed-book approach and standard RAG systems.
>
> **4. Refined Theoretical Analysis**
> We improve the Theoratical Guarantees subsection by adding the Curse of Dimensionality and Sample Complexity perspectives.

---

### Meta-Review · Area_Chair_wBp2 · 2025-12-29

**Summary:**

This paper proposes a modular distillation framework that decomposes knowledge-intensive reasoning into Analyzer, Informant, and Reasoner modules, aiming to transfer reasoning capabilities from large teacher models to smaller students. Reviewers generally find the idea interesting and practically motivated, with some evidence of gains on OBQA, StrategyQA, and MedQA.

However, there are several major unresolved concerns that limit confidence in the claims and overall contribution:
- Insufficient evaluation on harder reasoning tasks. The experimental validation is restricted to relatively narrow QA benchmarks. While the authors argue that datasets such as MATH or AIME are not “knowledge-intensive,” this distinction is not convincing: these tasks also require substantial factual knowledge, structured reasoning, and robustness to long reasoning chains. Without experiments on harder and more diverse benchmarks, it remains unclear whether the proposed modular framework generalizes beyond the chosen datasets or meaningfully advances reasoning capabilities.
- Limited model coverage. All student experiments are conducted on LLaMA-series models. There is no evaluation on other strong and widely used model families (e.g., Qwen), even at similar parameter scales. This raises concerns about whether the observed gains are architecture-specific rather than method-general.
- Strong structural assumptions and error propagation. The method relies on a fixed decomposition into subquestion generation, knowledge retrieval, and reasoning. This imposes a strong human prior on how reasoning should be structured, which reviewers note may not align with the diverse and adaptive reasoning patterns exhibited by modern LLMs. Moreover, the sequential pipeline is vulnerable to error propagation across modules, and the empirical analysis does not convincingly demonstrate robustness to upstream failures.

Overall, while the paper presents a clear idea, the lack of evaluation on harder tasks, the limited range of base models, and the strong, potentially non-general assumptions about reasoning structure significantly weaken the contribution. I therefore recommend rejection.

**Reviewer Concerns:**

The rebuttal partially addressed methodological clarification concerns raised by reviewers, particularly by adding the JRD baseline to disentangle architectural benefits from data enrichment, expanding theoretical justification for modular decomposition, and providing qualitative arguments and ablations to mitigate worries about error propagation (e.g., skip connections, fallback behavior, and robustness of the Reasoner). These additions reasonably respond to questions about whether gains stem from modularity itself and clarify the authors’ intended scope (knowledge-intensive QA under resource constraints).

However, the core concerns remain outstanding. The rebuttal does not add experiments on harder and more diverse reasoning benchmarks (e.g., MATH/AIME), does not evaluate on stronger or alternative base models (e.g., Qwen series), and does not empirically demonstrate robustness of the fixed Analyzer–Informant–Reasoner decomposition under more complex or diverse reasoning behaviors. As a result, doubts about generalization, architecture-specific effects, and the strength of the imposed human prior are not convincingly resolved.

**Reviewer Scores:**

Overall, scores would likely remain largely unchanged: reviewers with mixed or borderline scores might have made minor upward adjustments due to added baselines and clarifications, but reviewers primarily concerned with generalization, task difficulty, and model coverage would likely not increase their scores, i.e., still 2,4,6.

---

### Decision · Program_Chairs · 2026-01-26

Reject